# AUTOMATIC TRANSFORMATION OF IRREDUCIBLE REPRESENTATIONS FOR EFFICIENT CONTRACTION OF TENSORS WITH CYCLIC GROUP SYMMETRY

YANG GAO*, PHILLIP HELMS†, GARNET KIN-LIC CHAN‡, AND EDGAR SOLOMONIK§

**Abstract.** Tensor contractions are ubiquitous in computational chemistry and physics, where tensors generally represent states or operators and contractions express the algebra of these quantities. In this context, the states and operators often preserve physical conservation laws, which are manifested as group symmetries in the tensors. These group symmetries imply that each tensor has block sparsity and can be stored in a reduced form. For nontrivial contractions, the memory footprint and cost are lowered, respectively, by a linear and a quadratic factor in the number of symmetry sectors. State-of-the-art tensor contraction software libraries exploit this opportunity by iterating over blocks or using general block-sparse tensor representations. Both approaches entail overhead in performance and code complexity. With intuition aided by tensor diagrams, we present a technique, irreducible representation alignment, which enables efficient handling of Abelian group symmetries via only dense tensors, by using contraction-specific reduced forms. This technique yields a general algorithm for arbitrary group symmetric contractions, which we implement in Python and apply to a variety of representative contractions from quantum chemistry and tensor network methods. As a consequence of relying on only dense tensor contractions, we can easily make use of efficient batched matrix multiplication via Intel's MKL and distributed tensor contraction via the Cyclops library, achieving good efficiency and parallel scalability on up to 4096 Knights Landing cores of a supercomputer.

**1. Introduction.** Tensor contractions are computational primitives found in many areas of science, mathematics, and engineering. In this work, we describe how to accelerate tensor contractions involving block sparse tensors whose structure is induced by a cyclic group symmetry or a product of cyclic group symmetries. Tensors of this kind arise frequently in many applications, for example, in quantum simulations of many-body systems. By introducing a remapping of the tensor contraction, we show how such block sparse tensor operations can be expressed almost fully in terms of dense tensor operations. This approach enables effective parallelization and makes it easier to achieve peak performance by avoiding the complications of managing block sparsity. We illustrate the performance and scalability of our approach by numerical examples drawn from the contractions used in tensor network algorithms and coupled cluster theory, two widely used methods of quantum simulation.

A tensor $\mathcal{T}$ is defined by a set of real or complex numbers indexed by tuples of integers (indices) $i, j, k, l, \ldots$, where the indices take integer values $i \in 1 \ldots D_i, j \in 1 \ldots D_j, \ldots$ etc., and a single tensor element is denoted $t_{ijkl\ldots}$. We refer to the number of indices of the tensor as its *order* and the sizes of their ranges as its *dimensions* $(D_i \times D_j \times \cdots)$. We call the set of indices *modes* of the tensor. Tensor contractions are represented by a sum over indices of two tensors. In the case of matrices and vectors, the only possible contractions correspond to matrix and vector products. For higher order tensors, there are more possibilities, and an example of a contraction of two order 4 tensors is

$$w_{abij} = \sum_{k,l} u_{abkl} v_{klij}. \tag{1.1}$$

To illustrate the structure of the contraction, it is convenient to employ a graphical notation where a tensor is a vertex with each incident line representing a mode, and contracted modes are represented by lines joining vertices, as shown in Figure 1. Tensor contractions can be reduced to matrix multiplication (or a simpler matrix/vector operation) after appropriate transposition of the data to interchange the order of modes.

*Division of Engineering and Applied Science, California Institute of Technology (ygao@caltech.edu)
†Division of Chemistry and Chemical Engineering, California Institute of Technology (phelms@caltech.edu)
‡Division of Chemistry and Chemical Engineering, California Institute of Technology (garnetc@caltech.edu)
§Department of Computer Science, University of Illinois at Urbana-Champaign (solomonik@cs.illinois.edu)

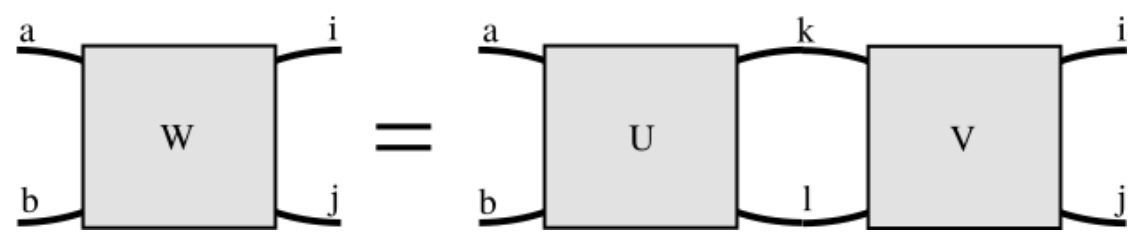

FIGURE 1. *Representation of the contraction in Eq. (1.1). Each tensor is represented by a vertex and each mode by a line; the lines joining the vertices are contracted over.*

In many physics and chemistry applications, there is an underlying symmetry group which constrains the relevant computations. This implies that under the operations of the group, the computational objects (e.g. the tensors) are transformed by a matrix representation of the group, which can be decomposed into irreducible representations (irreps) of the group. Computationally, the elements of the tensors are thus constrained, and each tensor can be stored in a compressed form, referred to as its *reduced form*. A special structure that often appears is one that is associated with a cyclic group. If each index transforms as an irrep of such a group and the overall tensor transforms as the symmetric representation, this constraint can be satisfied by a sparsity structure defined on the indices, e.g.

$$t_{\mathbf{ijk}\ldots} = 0 \quad \text{if} \quad \lfloor \mathbf{i}/G_1 \rfloor + \lfloor \mathbf{j}/G_2 \rfloor + \lfloor \mathbf{k}/G_3 \rfloor + \cdots \neq 0 \pmod{G}, \tag{1.2}$$

where the offset $G_i$ denotes the size of the symmetry group for the index $i$.

For a matrix, such sparsity would lead to a blocked matrix where each block is the same size, $G_1 \times G_2$. The blocks of an order 3 tensor would similarly all have the same dimensions, $G_1 \times G_2 \times G_3$. We refer to such tensors as tensors with *cyclic group symmetry*, or cyclic group tensors for short. In some applications, the block sizes are non-uniform, but this can be accommodated in a cyclic group tensor by padding blocks with zeros to a fixed size during initialization. With this assumption, the original tensor indices can be unfolded into symmetry modes and the symmetry blocks, where the symmetry modes fully express the block sparse structure,

$$t_{iI,jJ,kK\ldots} = 0 \quad \text{if} \quad I + J + K \cdots \neq 0 \pmod{G},$$

where we use the convention that the uppercase indices are the symmetry modes and the lowercase letters index into the symmetry blocks. The relationship between the symmetry modes is referred to as a symmetry conservation rule.

Given a number of symmetry sectors $G$ (as in (1.2)), cyclic group symmetry can reduce tensor contraction cost by a factor of $G$ for some simple contractions and $G^2$ for most contractions of interest (any contraction with a cost that is superlinear in input/output size). State-of-the-art sequential and parallel libraries for handling cyclic group symmetry, both in specific physical applications and in domain-agnostic settings, typically iterate over non-zero blocks within a block-sparse tensor format [1, 4, 12, 16, 18–21, 28, 32, 34, 39]. The use of explicit looping (over possibly small blocks) makes it difficult to reach theoretical peak compute performance. Parallelization of block-wise contractions can be done manually or via specialized software [12, 16, 19–21, 28, 32, 34]. However, such parallelization is challenging in the distributed-memory setting, where block-wise multiplication might (depending on contraction and initial tensor data distribution) require communication/redistribution of tensor data.

We introduce a general transformation of cyclic group symmetric tensors, *irreducible representation alignment*, which allows all contractions between such tensors to be transformed into a single large dense tensor contraction with optimal cost, in which the two input reduced forms as well as the output are indexed by a new auxiliary index. This transformation provides three advantages:

1. it avoids the need for data structures to handle block sparsity or scheduling over blocks,
2. it makes possible an efficient software abstraction to contract tensors with cyclic group symmetry,
3. it enables effective use of parallel libraries for dense tensor contraction and batched matrix multiplication.

The most closely related previous work to our approach that we are aware of is the *direct product decomposition (DPD)* [24,43], which similarly seeks an aligned representation of the two tensor operands. However, the unfolded structure of cyclic group tensors in Eq. (1) allows for a much simpler conversion to an aligned representation, both conceptually and in terms of implementation complexity. In particular, our approach can be implemented efficiently with existing dense tensor contraction primitives.

We develop a software library, *Symtensor*, that implements the irrep alignment algorithm and contraction. We study the efficacy of this new method for tensor contractions with cyclic group symmetry arising in physics and chemistry applications. Specifically, we consider some of the costly contractions arising in tensor network (TN) methods for quantum many-body systems and in coupled cluster (CC) theory for electronic structure calculations. We demonstrate that across a variety of tensor contractions, the library achieves orders of magnitude improvements in parallel performance and a matching sequential performance relative to the manual loop-over-blocks approach. The resulting algorithm may also be easily and automatically parallelized for distributed-memory architectures. Using the Cyclops Tensor Framework (CTF) library [40] as the contraction backend to Symtensor, we demonstrate good strong and weak scalability with up to at least 4096 Knights Landing cores of the Stampede2 supercomputer.

**2. Irreducible Representation Alignment Algorithm.** We now describe our proposed approach. We first describe the algorithm on an example contraction and provide intuition for correctness based on conservation of flow in a tensor diagram graph. These arguments are analogous to the conservation arguments used in computations with Feynman diagrams (e.g. momentum and energy conservation) [9] or with quantum numbers in tensor networks [37], although the notation we use is slightly different. We then give an algebraic derivation of all steps, which allows for a concrete proof of correctness and explicit expression for the cost in the general case.

**2.1. Example of the Algorithm.** We consider a contraction of order 4 tensors $\boldsymbol{\mathcal{U}}$ and $\boldsymbol{\mathcal{V}}$ into a new order 4 tensor $\boldsymbol{\mathcal{W}}$, where all tensors have cyclic group symmetry. We can express this cyclic group symmetric contraction as a contraction of tensors of order 8, by separating indices into symmetry-block (lower-case) indices and symmetry-mode (upper-case) indices, so

$$w_{aA,bB,iI,jJ} = \sum_{k,K,l,L} u_{aA,bB,kK,lL} v_{kK,lL,iI,jJ}.$$

Here and later we use commas to separate index groups for readability. The input and output tensors are assumed to transform as symmetric irreps of a cyclic group, which implies the following relationships between the symmetry modes and associated block structures,

$$\begin{aligned} w_{aA,bB,iI,jJ} &\neq 0 \text{ if } A+B-I-J \equiv 0 \pmod{G}, \\ u_{aA,bB,kK,lL} &\neq 0 \text{ if } A+B-K-L \equiv 0 \pmod{G}, \\ v_{kK,lL,iI,jJ} &\neq 0 \text{ if } K+L-I-J \equiv 0 \pmod{G}. \end{aligned}$$

Ignoring the symmetry, this tensor contraction would have cost $O(n^4G^4)$ for memory footprint and $O(n^6G^6)$ for computation, where $n$ is the dimension of each symmetry sector.

With the use of symmetry, the cost for memory and computation can be reduced to $O(n^4G^3)$ and $O(n^6G^4)$ respectively. This can be achieved by first representing the original tensor in a reduced dense form indexed by just 3 symmetry modes. In particular, we refer to the reduced form indexed by 3 symmetry modes that are a subset of the symmetry modes of the original tensors, as the standard reduced forms. The equations below show the mapping from the original tensor to one of its standard reduced form. The associated graphical notation is shown in Figure 2. This follows the same notation as in Figure 1, but arrows are now used to indicate the sign associated with the symmetry mode in the symmetry conservation rule for the tensor.

$$\begin{aligned}
\bar{w}_{aA,b,iI,jJ} &= w_{aA,b,I+J-A \bmod G,iI,jJ},\\
\bar{u}_{aA,b,kK,lL} &= u_{aA,b,K+L-A \bmod G,kK,lL},\\
\bar{v}_{kK,l,iI,jJ} &= v_{kK,l,I+J-K \bmod G,iI,jJ}.
\end{aligned}$$

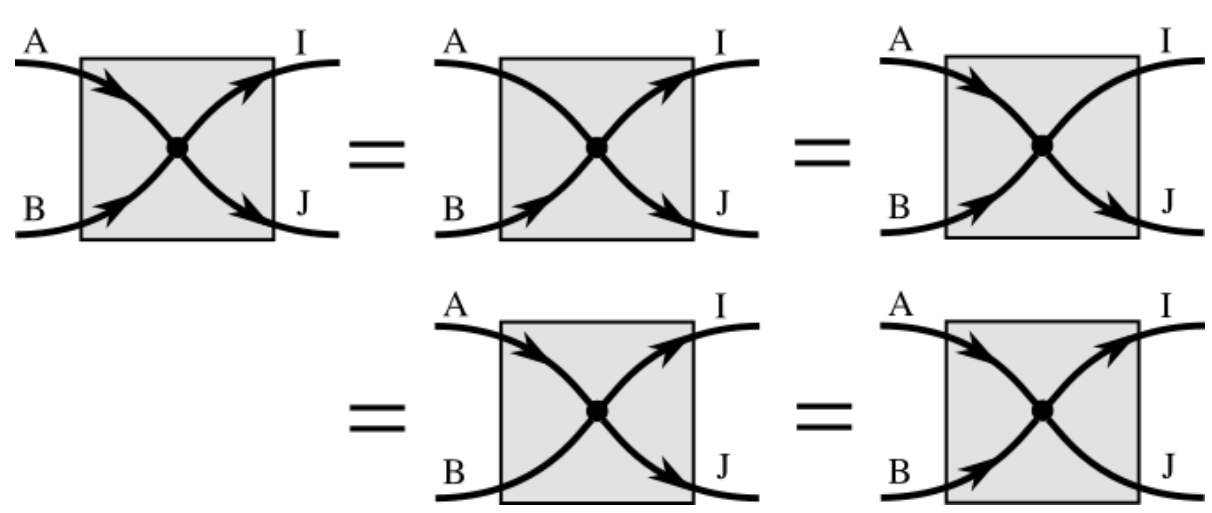

FIGURE 2. *Tensor diagrams of the standard reduced form. Arrows on each leg represent the corresponding symmetry indices that are explicitly stored. Symmetry indices on legs without arrows are not stored but are implicitly represented with the symmetry conservation law at the vertex* $(A + B = I + J \pmod G)$. *Note the lower-case indices for each symmetry blocks are always stored.*

The standard reduced form provides an implicit representation of the unstored symmetry mode due to symmetry conservation and can be easily used to implement the block-wise contraction approach prevalent in many libraries. This is achieved via manual loop nest over the appropriate symmetry modes of the input tensors, as shown in Algorithm 2.1. All elements of $\boldsymbol{\mathcal{W}}, \boldsymbol{\mathcal{U}}$, and $\boldsymbol{\mathcal{V}}$ in the standard reduced form can be accessed with 4 independent nested-loops to perform the multiply and accumulate operation in Algorithm 2.1. The other two implicit symmetry modes can be obtained inside these loops using symmetry conservation, reducing the computation cost to $O(G^4)$.

However, the indirection needed to compute $L$ and $J$ within the innermost loops prevents expression of the contraction in terms of standard library operations for a single contraction of dense tensors. Figure 3 illustrates that standard reduced forms cannot simply be contracted to obtain a reduced form as a result. The need to parallelize general block-wise tensor contraction operations in the nested loop approach above, creates a significant software-engineering challenge and computational overhead for tensor contraction libraries [16].

The main idea in the irreducible representation alignment algorithm is to first transform (reindex) the tensors using an auxiliary symmetry mode which subsequently allows a dense tensor contraction to be performed without the need for any indirection. In the above contraction, we define the auxiliary mode index as $Q \equiv I + J \equiv A + B \equiv K + L \pmod G$ and thus obtain a new reduced form for each tensor. The relations of this reduced form with

**Algorithm 2.1** Loop nest to perform group symmetric contraction $w_{aA,bB,iI,jJ} = \sum_{k,K,l,L} u_{aA,bB,kK,lL} v_{kK,lL,iI,jJ}$ using standard reduced forms $\bar{w}_{aA,bB,iI,j}$, $\bar{u}_{aA,bB,kK,l}$, and $\bar{v}_{kK,lL,iI,j}$.

**for** $A = 1, \dots, G$ **do**
  **for** $B = 1, \dots, G$ **do**
    **for** $I = 1, \dots, G$ **do**
      $J = A + B - I \bmod G$
      **for** $K = 1, \dots, G$ **do**
        $L = A + B - K \bmod G$
        $\forall a, b, i, j, \quad \bar{w}_{aA,bB,iI,j} = \bar{w}_{aA,bB,iI,j} + \sum_{k,l} \bar{u}_{aA,bB,kK,l} \bar{v}_{kK,lL,iI,j}$
      **end for**
    **end for**
  **end for**
**end for**

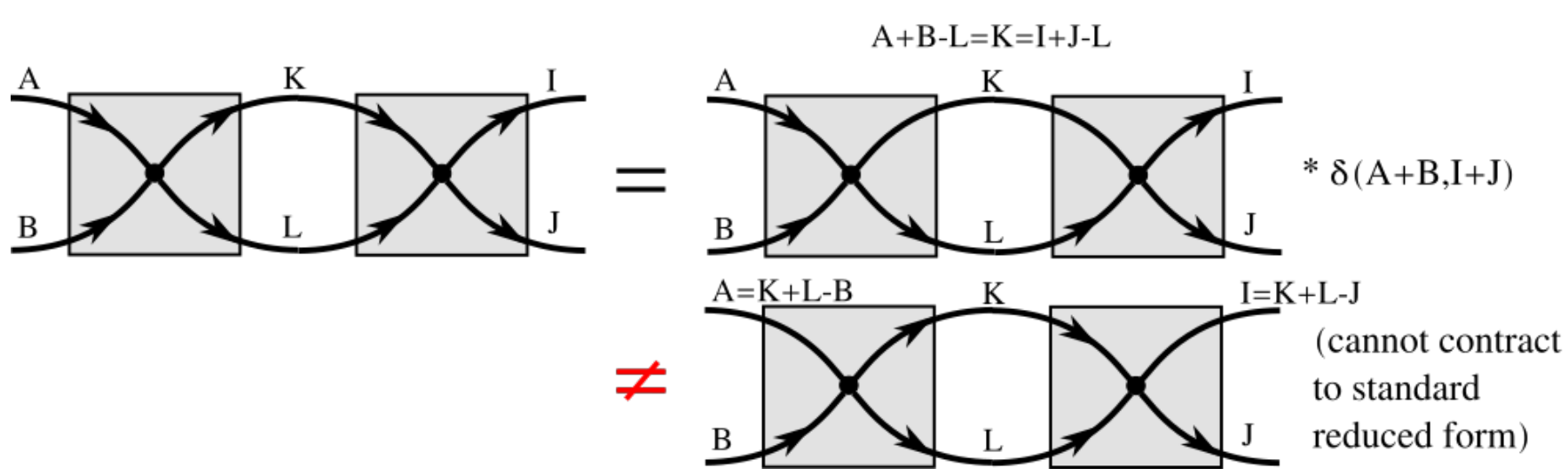

FIGURE 3. *These two tensor diagram equations aim to illustrate why certain reduced forms cannot be contracted directly. With the reduced forms chosen in the top case, the implicitly represented symmetry mode $K$ is not matched between the two input tensors and the output thus violates the true symmetry conservation rule unless a multiplication with a Kronecker delta tensor is performed. Even so, this contraction comes with an unfavorable scaling of $O(G^5)$. In the second case, the reduced forms cannot be contracted to produce a valid standard reduced form for the output (one needs 3 of the uncontracted indices to be represented / marked with arrows).*

the sparse form are as follows:

$$\hat{w}_{aA,b,i,jJ,Q} = w_{aA,b,Q-A \bmod G,i,Q-J \bmod G,jJ},$$
$$\hat{u}_{aA,b,k,lL,Q} = u_{aA,b,Q-A \bmod G,k,Q-L \bmod G,lL},$$
$$\hat{v}_{k,lL,i,jJ,Q} = v_{k,Q-L \bmod G,lL,i,Q-J \bmod G,jJ}.$$

This reduced form is displayed in Figure 4.

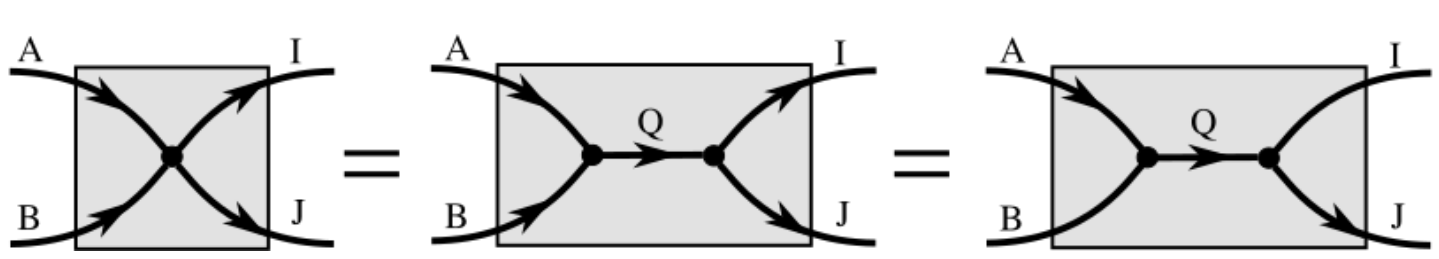

FIGURE 4. *The symmetry aligned reduced form is defined by introducing the $Q$ symmetry mode. Each of the two vertices defines a symmetry conservation relation: $A + B = Q \pmod G$ and $Q = I + J \pmod G$, allowing two of the arrows to be removed in the 3rd diagram, i.e. to be represented implicitly as opposed to being part of the reduced form.*

The $Q$ symmetry mode is chosen so that it can serve as part of the reduced forms of each of $\mathcal{U}$, $\mathcal{V}$, and $\mathcal{W}$. An intuition for why this alignment is possible is given via tensor diagrams

in Figure 5. The new auxiliary indices ($P$ and $Q$) of the two contracted tensors satisfy a conservation law $P = Q$, and so can be reduced to a single index.

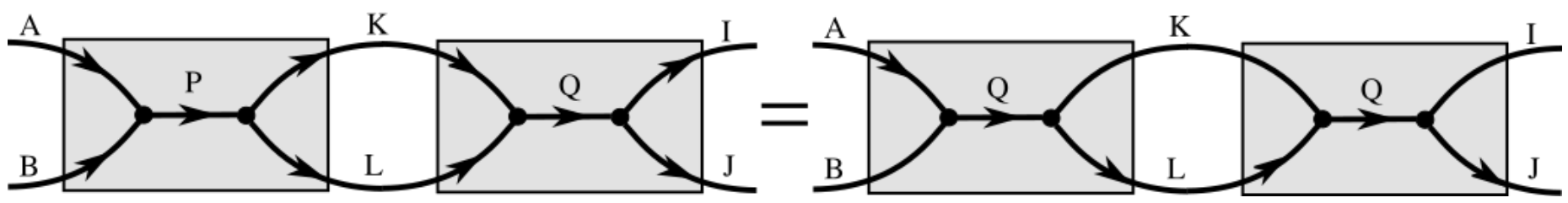

FIGURE 5. *By defining conservation laws on the vertices, we see that $P = K + L \pmod{G}$ and $K + L = Q \pmod{G}$. Consequently, the only non-zero contributions to the contraction must have $P = Q$.*

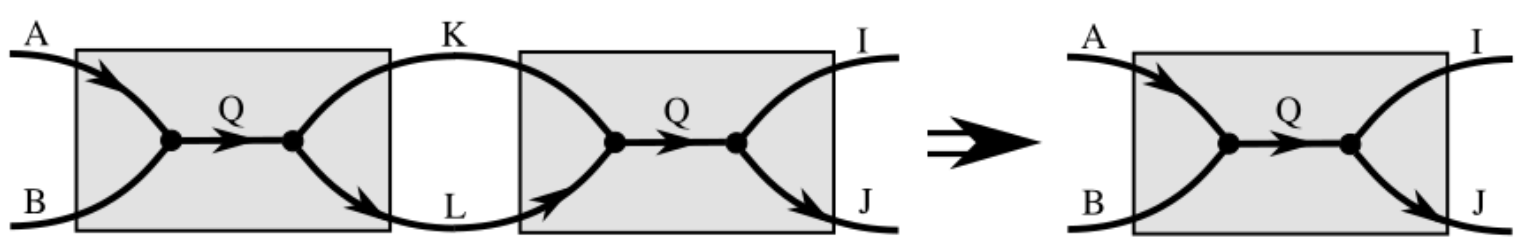

FIGURE 6. *The reduced forms may be contracted efficiently to produce the output reduced form. Ignoring intra-block indices, the resulting contraction may be performed with the `einsum` operation `W=einsum("AQL,LQJ->AQJ",U,V)`.*

As shown in Figure 6, given the aligned reduced forms of the two operands, we can contract them directly to obtain a reduced form for the output that also has the additional symmetry mode $Q$. Specifically, it suffices to perform the dense tensor contraction,

$$\hat{w}_{aA,b,i,jJ,Q} = \sum_{L,k,l} \hat{u}_{aA,b,k,lL,Q} \hat{v}_{k,lL,i,jJ,Q}.$$

This contraction can be expressed as a single `einsum` operation (available via NumPy, CTF, etc.) and can be done via a batched matrix multiplication (available in Intel's MKL). Once $\hat{\mathcal{W}}$ is obtained in this reduced form, it can be remapped to any other desired reduced form.

The remaining step is to define how to carry out the transformations between the aligned reduced forms and the standard reduced form. These can be performed via contraction with a Kronecker delta tensor defined on the symmetry modes, constructed from symmetry conservation, e.g, $\hat{u}_{aA,b,k,lL,Q} = \sum_B \bar{u}_{aA,bB,IL,j} \delta_{A,B,Q}$, where

$$\delta_{A,B,Q} = 0 \quad \text{if} \quad A + B - Q \neq 0 \pmod{G}. \tag{2.1}$$

Using this approach, all steps in our algorithm can be expressed fully in terms of single dense, or batched dense, tensor contractions.

**2.2. Generalization to Higher-Order Tensors.** We now describe how to generalize the algorithm to tensors of arbitrary order, including the more general symmetry conservation rules. We represent an order $N$ complex tensor with cyclic group symmetry as in (1.2) as an order $2N$ tensor, $\boldsymbol{\mathcal{T}} \in \mathbb{C}^{n_1 \times H_1 \times \cdots \times n_N \times H_N}$ satisfying, modulus remainder $Z \in \{1 \ldots G\}$ for coefficients $c_1 \ldots c_N$ with $c_i = G/H_i$ or $c_i = -G/H_i$,

$$t_{i_1 I_1 \ldots i_N I_N} = \begin{cases} r^{(T)}_{i_1 I_1 \ldots i_N I_N} & : c_1 I_1 + \cdots + c_N I_N \equiv Z \pmod{G} \\ 0 & : \text{otherwise}, \end{cases} \tag{2.2}$$

where the order $2N - 1$ tensor $\boldsymbol{\mathcal{R}}^{(T)}$ is the *reduced form* of the cyclic group tensor $\boldsymbol{\mathcal{T}}$. For example, the symmetry conservation rules in the previous section follow Eq. (2.2) with coefficients that are either $1$ or $-1$ ($G = H_i$).

Any cyclic group symmetry may be more generally expressed using a generalized Kronecker delta tensor with binary values, $\boldsymbol{\delta}^{(T)} \in \{0,1\}^{H_1 \times \cdots \times H_N}$ as

$$t_{i_1 I_1 \ldots i_N I_N} = r^{(T)}_{i_1 I_1 \ldots i_N I_N} \delta^{(T)}_{I_1 \ldots I_N}. \tag{2.3}$$

Specifically, the elements of the generalized Kronecker delta tensor are defined by

$$\delta^{(T)}_{I_1 \ldots I_N} = \begin{cases} 1 & : c_1 I_1 + \cdots + c_N I_N \equiv Z \pmod{G} \\ 0 & : \text{otherwise.} \end{cases}$$

Using these generalized Kronecker delta tensors, we provide a specification of our approach for arbitrary tensor contractions (Figure 7) in Algorithm 2.2. This algorithm performs any contraction of two tensors with cyclic group symmetry, written for some $s, t, v \in \{0, 1, \ldots\}$, as

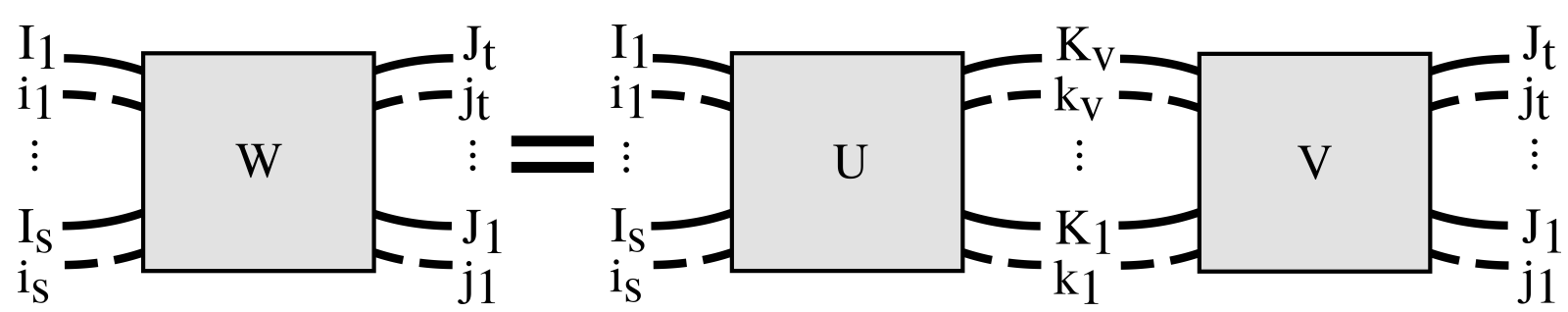

FIGURE 7. *A contraction of a tensor of order $s+v$ with a tensor of order $v+t$ into a tensor of order $s+t$, where all tensors have cyclic group symmetry and are represented with tensors of twice the order. Note that unlike in the previous section, the lines are not labelled by arrows (denoting coefficients $1$ or $-1$), but are associated with more general integer coefficients $c_i = \pm G/H_i$, to give symmetry conservation rules of the form Eq. (2.2).*

$$w_{i_1 I_1 \ldots i_s I_s j_1 J_1 \ldots j_t J_t} = \sum_{k_1 K_1 \ldots k_v K_v} u_{i_1 I_1 \ldots i_s I_s k_1 K_1 \ldots k_v K_v} v_{k_1 K_1 \ldots k_v K_v j_1 J_1 \ldots j_t J_t}. \tag{2.4}$$

The algorithm assumes the coefficients defining the symmetry of $\boldsymbol{\mathcal{U}}$ and $\boldsymbol{\mathcal{V}}$ match for the indices $K_1 \ldots K_v$ (it is also easy to allow for the coefficients to differ by a sign, as is the case in the contraction considered in Section 2.1).

Algorithm 2.2 also details the cost of each step. As opposed to the $O((nG)^{s+t+v})$ cost of the naive approach which does not use symmetry, Algorithm 2.2 achieves an overall arithmetic cost of

$$O\Big(((nG)^{s+v} + (nG)^{v+t} + (nG)^{s+t})/G + (nG)^{s+t+v}/G^2\Big).$$

Achieving this cost relies on obtaining the desired reduced forms by implicitly contracting with generalized Kronecker deltas (reordering and rescaling tensor elements) as opposed to the cost of treating the transformation as a general (dense) tensor contraction. The latter would entail a cost that is greater overall by a factor of $O(G)$ when $s$, $t$, or $v$ is 0.

**2.3. Algebraic Proof of Correctness.** We now provide a proof of correctness for Algorithm 2.2, which also serves as an alternate derivation of our method. Without loss of generality, we consider the case when $s, t, v = 2$, and ignore intra-block (lowercase) indices. In Algorithm 2.2, only one mode from the groups $(I_1, \ldots, I_s)$, $(J_1, \ldots, J_t)$, $(K_1, \ldots, K_v)$ is kept implicit at a time, so all other modes arising in the general case (arbitrary $s, t, v$) may be easily carried through the below derivation.

We now show that the generalized Kronecker delta tensors $\boldsymbol{\delta}^{(1)}$, $\boldsymbol{\delta}^{(2)}$, and $\boldsymbol{\delta}^{(3)}$ defined on line 4 of Algorithm 2.2 and the new reduced forms on line 5, may be derived from algebraic

**Algorithm 2.2** The irrep alignment algorithm for contraction of cyclic group symmetric tensors, for contraction defined as in (2.4).

1: Input two tensors $\boldsymbol{\mathcal{U}}$ of order $s+v$ and $\boldsymbol{\mathcal{V}}$ of order $v+t$ with symmetry conservation rules described using coefficient vectors $\boldsymbol{c}^{(U)}$ and $\boldsymbol{c}^{(V)}$ and remainders $Z^{(U)}$ and $Z^{(V)}$ as in (2.2).

2: Assume that these vectors share coefficients for contracted modes of the tensors, so that if $\boldsymbol{c}^{(U)} = \begin{bmatrix} \boldsymbol{c}_1^{(U)} \\ \boldsymbol{c}_2^{(U)} \end{bmatrix}$, then $\boldsymbol{c}^{(V)} = \begin{bmatrix} \boldsymbol{c}_2^{(U)} \\ \boldsymbol{c}_2^{(V)} \end{bmatrix}$.

3: Define new coefficient vectors, $\boldsymbol{c}^{(A)} = \begin{bmatrix} \boldsymbol{c}_1^{(U)} \\ 1 \end{bmatrix}$, $\boldsymbol{c}^{(B)} = \begin{bmatrix} \boldsymbol{c}_2^{(U)} \\ -1 \end{bmatrix}$, and $\boldsymbol{c}^{(C)} = \begin{bmatrix} \boldsymbol{c}_2^{(V)} \\ 1 \end{bmatrix}$.

4: Define generalized Kronecker deltas $\boldsymbol{\delta}^{(1)}$, $\boldsymbol{\delta}^{(2)}$, and $\boldsymbol{\delta}^{(3)}$ respectively based on the coefficient vectors $\boldsymbol{c}^{(A)}$, $\boldsymbol{c}^{(B)}$, $\boldsymbol{c}^{(C)}$ and remainders $Z^{(U)}$, 0, $Z^{(V)}$.

5: Let $\bar{\boldsymbol{\mathcal{R}}}^{(U)}$ and $\bar{\boldsymbol{\mathcal{R}}}^{(V)}$ be the given reduced forms for $\boldsymbol{\mathcal{U}}$ and $\boldsymbol{\mathcal{V}}$ (based on the generalized Kronecker deltas $\boldsymbol{\delta}^{(U)}$ and $\boldsymbol{\delta}^{(V)}$). Assume the reduced forms $\bar{\boldsymbol{\mathcal{R}}}^{(U)}$ and $\bar{\boldsymbol{\mathcal{R}}}^{(V)}$ for $\boldsymbol{\mathcal{U}}$ and $\boldsymbol{\mathcal{V}}$ do not store the last symmetry mode (other cases are similar). Compute the following new reduced forms $\boldsymbol{\mathcal{R}}^{(U)}$ and $\boldsymbol{\mathcal{R}}^{(V)}$, via contractions:

$$r^{(U)}_{i_1 I_1 \dots i_{s-1} I_{s-1} i_s k_1 K_1 \dots k_{v-1} K_{v-1} k_v Q} = \sum_{I_s K_v} \bar{r}^{(U)}_{i_1 I_1 \dots i_s I_s k_1 K_1 \dots k_{v-1} K_{v-1} k_v} \delta^{(1)}_{I_1 \dots I_s Q} \delta^{(2)}_{K_1 \dots K_v Q},$$

$$r^{(V)}_{k_1 K_1 \dots k_{v-1} K_{v-1} k_v j_1 J_1 \dots j_{t-1} J_{t-1} j_t Q} = \sum_{K_v J_t} \bar{r}^{(V)}_{k_1 K_1 \dots k_v K_v j_1 J_1 \dots j_{t-1} J_{t-1} j_t} \delta^{(2)}_{K_1 \dots K_v Q} \delta^{(3)}_{J_1 \dots J_t Q}.$$

▷ *The above contractions can be done with constant work per element of $\boldsymbol{\mathcal{R}}^{(U)}$ and $\boldsymbol{\mathcal{R}}^{(V)}$, namely $O(n^{s+v} G^{s+v-1})$ and $O(n^{v+t} G^{v+t-1})$, or with a factor of $O(G)$ more if done as dense tensor contractions that ignore the structure of $\boldsymbol{\delta}^{(1)}$, $\boldsymbol{\delta}^{(2)}$, and $\boldsymbol{\delta}^{(3)}$.*

6: Compute

$$r^{(W)}_{i_1 I_1 \dots i_{s-1} I_{s-1} i_s J_1 J_1 \dots j_{t-1} J_{t-1} j_t Q} = \sum_{k_1 K_1 \dots k_{v-1} K_{v-1} k_v} r^{(U)}_{i_1 I_1 \dots i_{s-1} I_{s-1} i_s k_1 K_1 \dots k_{v-1} K_{v-1} k_v Q} r^{(V)}_{k_1 K_1 \dots k_{t-1} K_{v-1} k_v j_1 J_1 \dots j_{t-1} J_{t-1} j_t Q}$$

▷ *The above contraction has cost $O(n^{s+t+v} G^{s+t+v-2})$*

7: If a standard output reduced form is desired, for example with the last mode of $\boldsymbol{\mathcal{W}}$ stored implicitly, then compute

$$\bar{r}^{(W)}_{i_1 I_1 \dots i_s I_s j_1 J_1 \dots j_{t-1} J_t j_t} = \sum_{Q} r^{(W)}_{i_1 I_1 \dots i_{s-1} I_{s-1} i_s J_1 J_1 \dots j_{t-1} J_{t-1} j_t Q} \delta^{(1)}_{I_1 \dots I_s Q}.$$

If we instead desire a reduced form with another implicit mode, it would not be implicit in $\boldsymbol{\mathcal{R}}^{(W)}$, so we would need to also contract with $\delta^{(3)}_{J_1 \dots J_t Q}$ and sum over the desired implicit mode.

▷ *In either case, the above contraction can be done with constant work per element of $\boldsymbol{\mathcal{R}}^{(W)}$, namely $O(n^{s+t} G^{s+t-1})$, or with a factor of $O(G)$ more if done as dense tensor contractions, if ignoring the structure of $\boldsymbol{\delta}^{(1)}$ and $\boldsymbol{\delta}^{(3)}$.*

manipulation of the contraction expressed in a standard reduced form. Using the standard reduced forms for the input tensors, we consider a refactorization of the generalized Kronecker delta tensors in the contraction,

$$w_{ABIJ} = \sum_{KL} \bar{r}^{(U)}_{ABK} \underbrace{\delta^{(U)}_{ABKL}\delta^{(V)}_{KLIJ}}_{\sum_Q \delta^{(1)}_{ABQ}\delta^{(2)}_{IJQ}\delta^{(3)}_{KLQ}} \bar{r}^{(V)}_{KIJ}.$$

We show that such a refactorization exists. We have that $\delta^{(U)}_{ABKL}\delta^{(V)}_{KLIJ} = 1$ whenever both of the following are satisfied,

$$\begin{aligned} c^{(U)}_1 A + c^{(U)}_2 B + c^{(U)}_3 K + c^{(U)}_4 L &\equiv Z^{(U)} \mod G, \\ c^{(V)}_1 K + c^{(V)}_2 L + c^{(V)}_3 I + c^{(V)}_4 J &\equiv Z^{(V)} \mod G. \end{aligned}$$

Since the coefficients for indices $K$ and $L$, namely $c^{(U)}_3$, $c^{(U)}_4$ and $c^{(V)}_1$, $c^{(V)}_2$ must match, the above two statements are satisfied if and only if there exists a unique $Q \in \{1, \ldots, G\}$, with which the following three statements are all satisfied,

$$\begin{aligned} c^{(U)}_1 A + c^{(U)}_2 B + Q &\equiv Z^{(U)} \mod G, \\ c^{(V)}_1 K + c^{(V)}_2 L - Q &\equiv 0 \mod G, \\ c^{(V)}_3 I + c^{(V)}_4 J + Q &\equiv Z^{(V)} \mod G. \end{aligned}$$

We can thus define new generalized Kronecker delta tensors so that $\delta^{(1)}_{ABQ}\delta^{(2)}_{KLQ}\delta^{(3)}_{IJQ} = 1$ whenever the above statements hold. These definitions match those specified via coefficients on line 3 of Algorithm 2.2.

With the $Q$ index defined from the refactorization, we define the symmetry-aligned reduced forms used in Algorithm 2.2. For $\boldsymbol{\mathcal{U}}$, this reduced form is given by $r^{(U)}_{AKQ} = u_{ABKL}$ whenever $c^{(U)}_1 A + c^{(U)}_2 B + Q \equiv Z^{(U)} \mod G$ and $c^{(V)}_1 K + c^{(V)}_2 L - Q \equiv 0 \mod G$. The new reduced form then satisfies,

$$r^{U}_{AKQ}\delta^{(1)}_{ABQ} = \bar{r}^{(U)}_{ABK}\delta^{(1)}_{ABQ},$$

and similarly for $\boldsymbol{\mathcal{V}}$. This equality allows us to perform the desired substitutions,

$$w_{ABIJ} = \sum_{K,L}\sum_{Q} \delta^{(1)}_{ABQ}\delta^{(2)}_{KLQ}\delta^{(3)}_{IJQ} r^{(U)}_{AKQ} r^{(V)}_{KIQ}.$$

After substituting the symmetry-aligned reduced forms, the two generalized Kronecker delta tensors defining the symmetry of the output may be factored out, while the third (associated with contracted modes) may be summed out,

$$w_{ABIJ} = \sum_{Q} \delta^{(1)}_{ABQ}\delta^{(3)}_{IJQ} \underbrace{\sum_{K} r^{(U)}_{AKQ} r^{(V)}_{KIQ}}_{r^{(W)}_{AIQ}}.$$

A reduced form for the result ($\boldsymbol{\mathcal{R}}^{(W)}$) is thus obtained from a contraction (line 6 of Algorithm 2.2) with $O(G^4)$ cost complexity.

```
import numpy as np
from symtensor import array, einsum

# Define Z3 Symmetry
irreps = [0,1,2]
G = 3
total_irrep = 0
z3sym  = ["++--", [irreps]*4, total_irrep, G]

# Initialize two sparse tensors as input
N = 10
Aarray = np.random.random([G,G,G,N,N,N,N])
Barray = np.random.random([G,G,G,N,N,N,N])

# Initialize symtensor with raw data and symmetry
u = array(Aarray, z3sym)
v = array(Barray, z3sym)

# Compute output symtensor
w = einsum('abkl,klij->abij', u,  v)
```

FIGURE 8. *Symtensor library example for contraction of two group symmetric tensors.*

**3. Library implementation.** We implement the irrep alignment algorithm as a Python library, Symtensor[1]. The library implements the algorithm described in Section 2, automatically selecting the appropriate reduced form to align the irreps for the contraction, constructing the generalized Kronecker deltas to convert input and output tensors to the target forms, and performing the batched dense tensor contractions that implement the numerical computation. The dense tensor contraction is interfaced to different contraction backends. Besides the default NumPy `einsum` backend, we also provide a backend that leverages MKL's batched matrix-multiplication routines [2] to obtain good threaded performance, and employ an interface to Cyclops [40] for distributed-memory execution.

In Figure 8, we provide an example of how the Symtensor library can be used to perform the contraction of two cyclic group tensors with $Z_3$ (cyclic group with $G = 3$) symmetry for each index. In the code, the Symtensor library initializes the order $4$ cyclic group symmetric tensor using an underlying order 7 dense reduced representation. Once the tensors are initialized, the subsequent `einsum` operation implements the contraction shown in Fig. 5 without referring to any symmetry information in its interface. While the example is based on a simple cyclic group for an order 4 tensor, the library supports arbitrary orders, as well as products of cyclic groups and infinite cyclic groups (e.g. $U(1)$ symmetries).

As introduced in Section 2, the main operations in our irrep alignment algorithm consist of transformation of the reduced form and the contraction of reduced forms. Symtensor chooses the least costly version of the irrep alignment algorithm from a space of variants defined by different choices of the implicitly represented modes of the three tensors in symmetry aligned reduced form in Algorithm 2.2 (therein these are the symmetry modes $I_s$, $J_t$, and $K_v$). This choice is made by enumerating all valid variants. After choosing the best reduced form, the required generalized Kronecker deltas, $\boldsymbol{\delta}^{(U,I)}$ and $\boldsymbol{\delta}^{(V,J)}$ in Algorithm 2.2, are generated as dense tensors. This permits both the transformations and the reduced form contraction to be done as `einsum` operations of dense tensors with the desired backend.

[1] https://github.com/yangcal/symtensor

**4. Example Applications.** As a testbed for this approach, we survey a few group symmetric tensor contractions that arise in computational quantum chemistry and quantum many-body physics methods. The emergence of cyclic group symmetric tensors in this context is attributable to symmetries such as those associated with the conservation of particle number, spin, and invariance to spatial transformations associated with point group and lattice symmetries [5, 22, 44]. Many numerical implementations in these fields leverage cyclic group symmetries [1,13,16,18,19,21,25,28,30,32,34,36,38,43], often via block sparse tensor formats. As described in Section 2.1, our proposed algorithm achieves the same computational improvement via transforming the cyclic group tensor representation, while maintaining a global view of the problem as a dense tensor contraction, as opposed to a series of block-wise operations or a contraction of block-sparse tensors.In this section, we introduce a few group symmetric tensor contractions that are costly components of a common quantum chemistry method (coupled cluster theory) and a common quantum many-body physics technique (tensor network simulations) and provide relevant background. These contractions, summarized in Table 1, are evaluated as part of a benchmark suite in Section 5 (we also consider a suite of synthetic contractions with different tensor order, i.e. different choices of $s, t, v$).

**4.1. Periodic Coupled Cluster Contractions.** When computing the electronic structure of molecules and materials, coupled cluster theories utilize tensor-based approximate representations of quantum wavefunctions [3, 6, 11, 45, 46], where more accurate representations require higher order tensors [17, 29, 35, 42]. Point group symmetries of the molecular structure, such as those associated with a rotational group $C_{nv}$ or a product group such as $D_{2h}$, usually provide the largest symmetry-related computational cost reductions for molecular systems. In crystalline (periodic) materials, the invariance of the atomic lattice to translation operations that are multiples of lattice vectors defines the crystal translational symmetry group, which is a product of cyclic symmetry groups along each lattice dimension. For a three-dimensional crystal, the size of the resulting symmetry group takes the form $G = G_1 \times G_2 \times G_3$, and $G_1$, $G_2$, and $G_3$ are usually called the number of $k$ points along each dimension. These are typically taken to be as large as computationally feasible, thus the savings arising from efficient use of crystal translational symmetry are particularly important in materials simulation [10, 14, 15, 25, 27].

Within periodic coupled cluster theory, three common expensive tensor contractions can be written as

$$
\begin{aligned}
w_{iI,jJ,mM,nN} &= \sum_{kK,lL} u_{iI,jJ,kK,lL} v_{mM,nN,kK,lL}, \\
w_{iI,jJ,mM,kK} &= \sum_{oO,pP} u_{oO,pP,iI,jJ} v_{oO,pP,kK,mM}, \\
w_{iI,jJ,mM,nN} &= \sum_{oO,pP} u_{oO,pP,mM,nN} v_{oO,pP,iI,jJ}.
\end{aligned}
$$

Each symmetry mode of the tensors is associated with the aforementioned crystal translational symmetry group. Physically, the indices fall into two classes, with modes $i, j, k, l$ and $m, n, o, p$ respectively called the virtual and occupied indices, each associated with a different dimension, however, we will for simplicity not distinguish between them and simply refer to their dimension as $N_i, N_j, \ldots$. If we assume all dimensions are the same, then the cost of the above contractions using cyclic group symmetry scales as $G^4 N^6$. These contractions are summarized in Table 1.

**4.2. Tensor Network Contractions.** In the context of quantum many-body physics, tensor networks provide a compact representation of a quantum state or operator as a con-

traction of many tensors. For one-dimensional and two-dimensional TN representations on a regular lattice, each lattice site is represented by a tensor with one index corresponding to the local state of the system and additional indices connecting it to nearest-neighbor tensors. These TNs are respectively called a matrix product state (MPS) [8, 31, 48] and projected entangled pair state (PEPS) [47] and are associated with many famous algorithms for computing quantum states such as the density matrix renormalization group [48, 49]. In the presence of certain (Abelian) physical symmetries such as the conservation of particle number, the tensors in the tensor network can be chosen to be cyclic group symmetric tensors [4, 26, 36, 49].

We consider two expensive TN contractions, each arising respectively from MPS or PEPS algorithms which aim to find the ground state of quantum many-body systems. From the MPS algorithm, we consider the contraction,

$$w_{iI,jJ,lL,mM} = \sum_{kK} u_{iI,jJ,kK} v_{kK,lL,mM},$$

which is encountered within an iterative eigensolver used to optimize a single MPS tensor. The cost of this contraction using cyclic group symmetries scales as $G^3 N_i N_j N_k N_j N_l$ with each index having the same number of symmetry blocks. In Table 1, we set $N = N_i = N_j = N_k = N_l = N_m$ for simplicity.

We consider the contraction with the highest scaling cost with respect to the dimension of the PEPS tensors,

$$w_{iI,jJ,mM,nN} = \sum_{kK,lL} u_{iI,jJ,kK,lL} v_{kK,lL,mM,nN},$$

where $\boldsymbol{\mathcal{U}}$ is part of the MPS and $\boldsymbol{\mathcal{V}}$ is a PEPS tensor. The PEPS contraction considered arises during the computation of the normalization of the quantum state, done here using an implicit version of the boundary contracton approach [33].

In the above two contractions, indices $i$, $j$ and $k$, $l$, $m$, $n$ connect tensors within the MPS and PEPS respectively. Each index has the same number of symmetry sectors $G$ and we set all indices connecting MPS and PEPS tensors to be equal, i.e. $N_{mps} = N_i = N_j$ and $N_{peps} = N_k = N_l = N_m = N_n$. This gives an overall cost of $G^4 \left(N_{mps}\right)^2 \left(N_{peps}\right)^4$. In Table 1, we set $N = N_{mps} = N_{peps}$ for simplicity.

**5. Performance Evaluation.** Performance experiments were carried out on the Stampede2 supercomputer. Each Stampede2 node is a Intel Knight's Landing (KNL) processor, on which we use up to 64 of 68 cores by employing up to 64 threads with single-node NumPy/MKL and 64 MPI processes per node with 1 thread per process with Cyclops. We use the Symtensor library together with one of three external contraction backends: Cyclops, default NumPy, or a batched BLAS backend for NumPy arrays (this backend leverages HPTT [41] for fast tensor transposition and dispatches to Intel's MKL BLAS for batched matrix multiplication).[2] We also compare against the loop over symmetry blocks algorithm as illustrated in Algorithm 2.1. This implementation performs each block-wise contraction using MKL, matching state of the art libraries for tensor contractions with cyclic group symmetry [7, 23].

**5.1. Single-Node Performance Results.** We consider the performance of Symtensor on a single core and a single node of KNL relative to manually-implemented looping over blocks as well as relative to naive contractions that ignore symmetry. Our manual loop implementation of contractions stores a Python list of NumPy arrays to represent the tensor blocks and invokes the NumPy `einsum` functions to perform each block-wise contraction.

[2]We use the default Intel compilers on Stampede2 with the following software versions: HPTT v1.0.0, CTF v1.5.5 (compiled with optimization flags: -O2 -no-ipo), and MKL v2018.0.2.

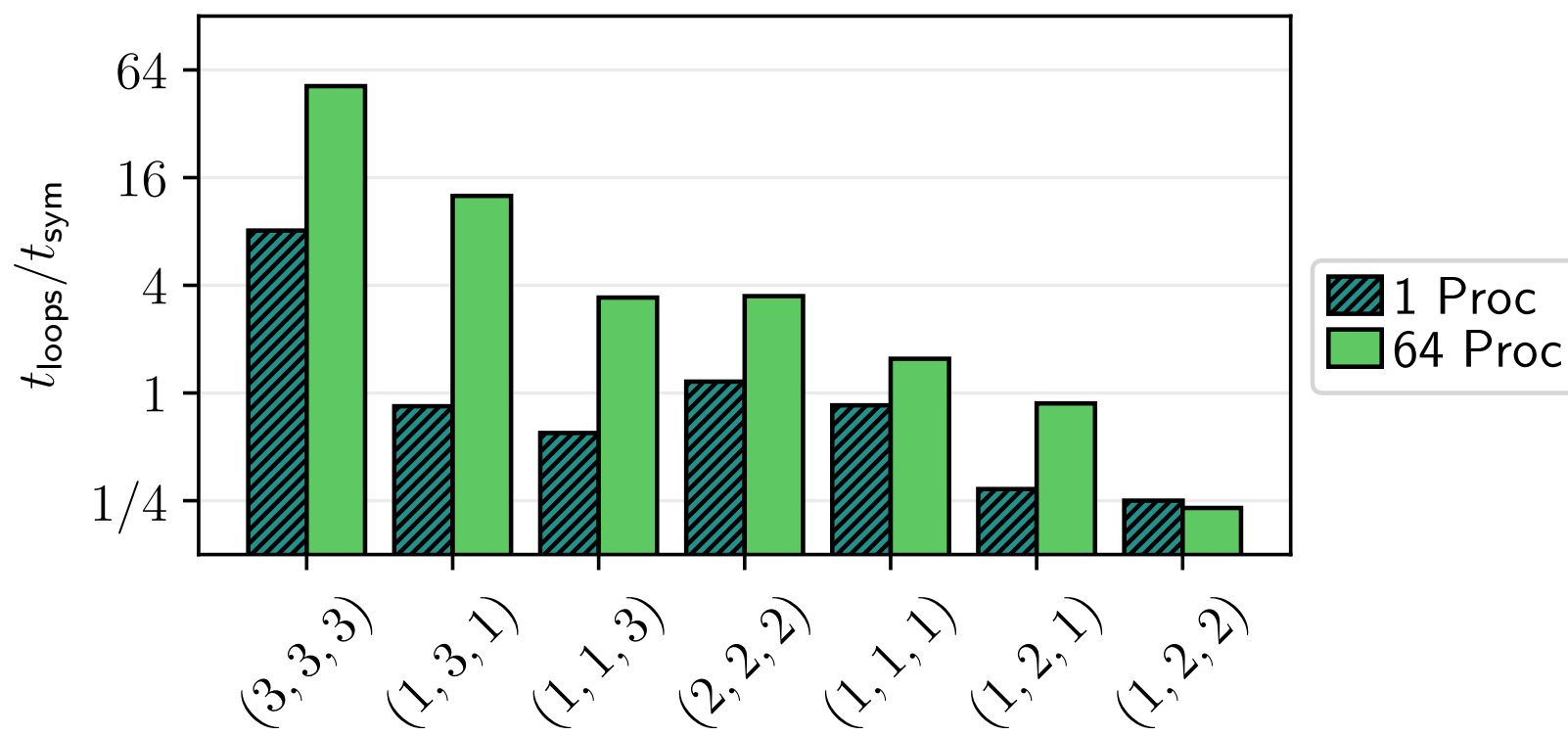

FIGURE 9. *Comparison of execution times for various types of contraction using the Symtensor library with batched BLAS and using loops over blocks with NumPy as the contraction backend. Results are shown for various combinations of (s, t, v) where the number of modes shared between the first input and output is s, between the second input and output is t, and between the two inputs (contracted) is v.*

**5.1.1. Sensitivity to Contraction Type.** We first examine the performance of the irrep alignment algorithm for generic contractions (Eq. 2.4) with equal dimensions $N$ and $G$ for each mode, but different ($s$, $t$, $v$). Figure 9 shows the speed-up in execution time obtained by Symtensor relative to the manual loop implementation. The contractions are constructed by fixing $G = 4$ and modifying $N$ to attain a fixed total of $80 \times 10^9$ floating-point operations. All contractions are performed on both a single thread and 64 threads of a single KNL node and timings are compared in those respective configurations. The irrep alignment algorithm achieves better parallel scalability than block-wise contraction and can also be faster sequentially. However, we also observe that in cases when one tensor is larger than others (when $s, t, v$ are unequal) the irrep alignment approach can incur overhead relative to the manual loop implementation. Overhead can largely be attributed to the cost of transformations between reduced representations, which are done as dense tensor contractions with the generalized Kronecker delta tensors. An alternative transformation mechanism that utilizes the structure of the delta tensors would forgo this factor of $O(G)$ overhead, but the use of dense tensor contractions permits use of existing optimized kernels and easy parallelizability.

TABLE 1

*Summary of coupled cluster and tensor network contractions used to benchmark the symmetric tensor contraction scheme and their costs. G is the size of the symmetry group and N is the dimension of each mode. We include matrix multiplication (MM) as a point of reference. The three CC contractions described in Section 4.1 are labeled $CC_1$,$CC_2$ and $CC_3$ respectively.*

| Label | Contraction | Symmetric Cost | Naive Cost |
|---|---|---|---|
| MM | $w_{iI,kK} = \sum_{jJ} u_{iI,jJ} v_{jJ,kK}$ | $\mathcal{O}(GN^3)$ | $\mathcal{O}(G^3N^3)$ |
| $CC_1$ | $w_{iI,jJ,mM,nN} = \sum_{kK,lL} u_{iI,jJ,kK,lL} v_{mM,nN,kK,lL}$ | $\mathcal{O}(G^4N^6)$ | $\mathcal{O}(G^6N^6)$ |
| $CC_2$ | $w_{iI,jJ,mM,kK} = \sum_{oO,pP} u_{oO,pP,iI,jJ} v_{oO,pP,kK,mM}$ | $\mathcal{O}(G^4N^6)$ | $\mathcal{O}(G^6N^6)$ |
| $CC_3$ | $w_{iI,jJ,mM,nN} = \sum_{oO,pP} u_{oO,pP,mM,nN} v_{oO,pP,iI,jJ}$ | $\mathcal{O}(G^4N^6)$ | $\mathcal{O}(G^6N^6)$ |
| MPS | $w_{iI,jJ,lL,mM} = \sum_{kK} u_{iI,jJ,kK} v_{kK,lL,mM}$ | $\mathcal{O}(G^3N^5)$ | $\mathcal{O}(G^5N^5)$ |
| PEPS | $w_{iI,jJ,mM,nN} = \sum_{kK,lL} u_{iI,jJ,kK,lL} v_{kK,lL,mM,nN}$ | $\mathcal{O}(G^4N^6)$ | $\mathcal{O}(G^6N^6)$ |

**5.1.2. Sensitivity to Symmetry Group Size for Application-Specific Contractions.** The results are displayed in Figure 10 with the top, center, and bottom plots showing the scal-

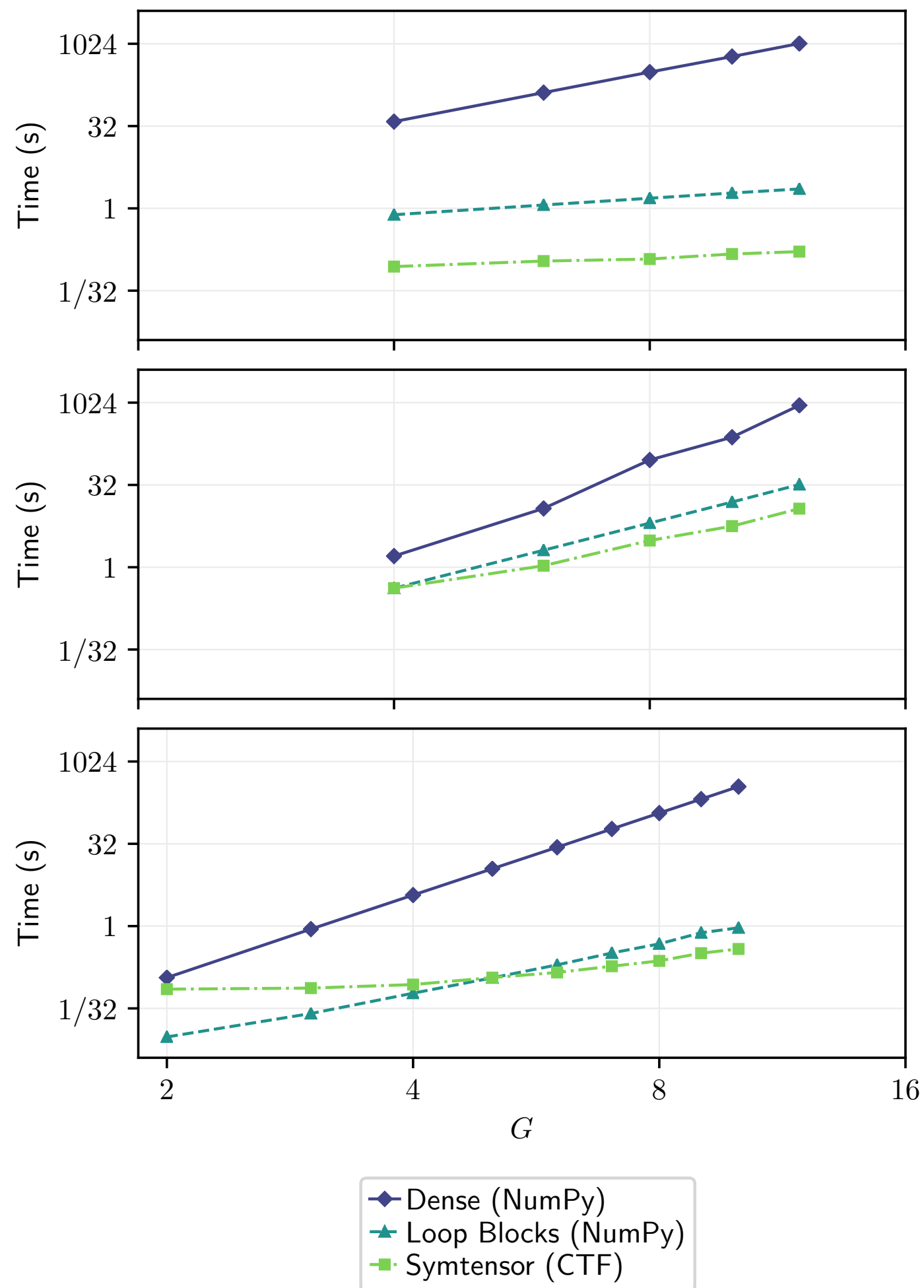

FIGURE 10. *Comparison of the execution times, in seconds, for contractions on a single thread using three different algorithms, namely a dense, non-symmetric contraction, loops over symmetry blocks, and our Symtensor library. From top to bottom, the plots show the scaling for matrix multiplication (MM), a coupled cluster contraction ($CC_1$), and a tensor network contraction (PEPS). The dense and loop over blocks calculations use NumPy as a contraction backend, while the Symtensor library here uses Cyclops as the contraction backend.*

ing for the contractions labeled MM, $CC_1$, and PEPS in Table 1. We compare scaling relative to two conventional approaches: a dense contraction without utilizing symmetry and loops over symmetry blocks, both using NumPy's `einsum` function. The dimensions of the tensors considered are, for matrix multiplication, $N = 500$ and $G \in [4, 12]$, for the CC contraction, $N_i = N_j = N_k = N_l = 8$, $N_a = N_b = N_c = N_d = 16$, with $G \in [4, 12]$, and for the PEPS contraction, $N_{\text{mps}} = 16$, $N_{\text{peps}} = 4$, with $G \in [2, 10]$. For all but the smallest contractions, using the Symtensor implementation improves contraction performance. A comparison of the slopes of the lines in each of the three plots indicates that the dense tensor contraction scheme results in a higher order asymptotic scaling of cost in $G$ than either of the symmetric approaches.

Figure 11 provides absolute performance with 1 thread and 64 threads for all contractions

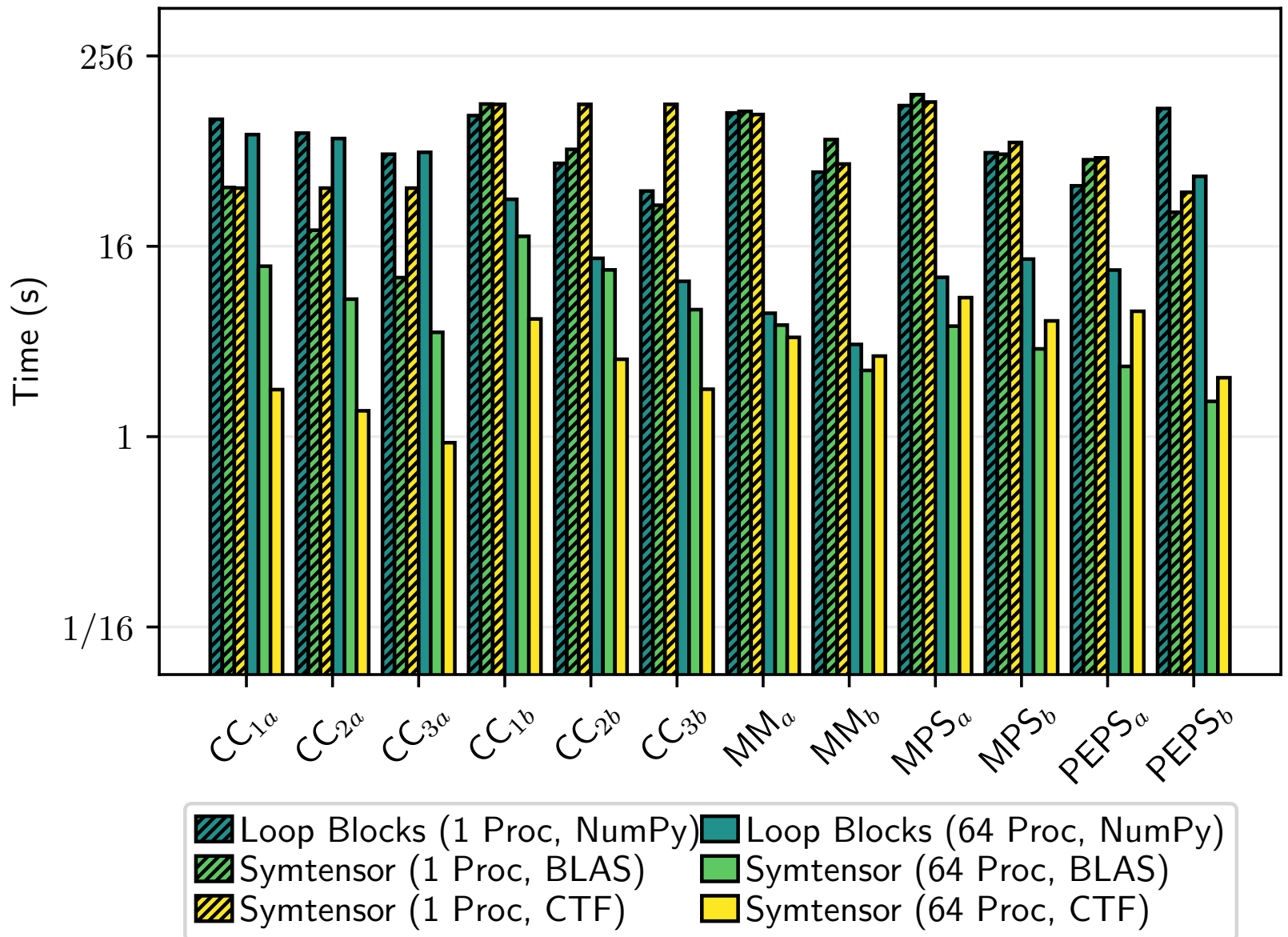

FIGURE 11. *Comparison of contraction times using the Symtensor library (using Cyclops for the array storage and contraction backend, or NumPy as the array storage with batched BLAS contraction backend) and loops over blocks using NumPy as the contraction backend. Results are shown for instances of the prototypical contractions introduced in Section 4, with details of tensor dimensions provided in Table 2. The different bars indicate both the algorithm and backend used and the number of threads used on a single node.*

in Table 1. For each contraction, we consider one with a large number of symmetry sectors ($G$) with small block size ($N$) (labeled with a subscript $a$) and another with fewer symmetry sectors and larger block size (labeled with a subscript $b$). The specific dimensions of all tensors studied are provided in Table 2. For each of these cases, we compare the execution time, in seconds, using loops over blocks dispatching to NumPy contractions, the Symtensor library with NumPy arrays and batched BLAS as the contraction backend, and the Symtensor library using Cyclops as the array and contraction backend.

A clear advantage in parallelizability of Symtensor is evident in Figure 11. With 64 threads, Symtensor outperforms manual looping by a factor of at least $1.4$X for all contraction benchmarks, and the largest speed-up, 69X, is obtained for the $CC_{3a}$ contraction. There is a significant difference between the contractions labeled to be of type $a$ (large $G$ and small $N$) and type $b$ (large $N$ and small $G$), with the geometric mean speedup for these two being $11$X and $2.8$X respectively on 64 threads; on a single thread, this difference is again observed, although less drastically, with respective geometric mean speedups of $1.9$X and $1.2$X. Type $b$ cases involve more symmetry blocks, amplifying overhead of manual looping.

**5.2. Multi-Node Performance Results.** We now illustrate the parallelizability of the irrep alignment algorithm by studying scalability across multiple nodes with distributed memory. All parallelization in Symtensor is handled via the Cyclops library in this case. The solid lines in Figure 12 show the strong scaling (fixed problem size) behavior of the Symtensor implementation on up to eight nodes of Stampede2. As a reference, we provide comparison to strong scaling on a single node for the loop over blocks method using NumPy as the array and contraction backend. We again observe that the Symtensor irrep alignment implementation provides a significant speedup over the loop over blocks strategy, which is especially evident

TABLE 2
*Dimensions of the tensors used for contractions in Figure 11 and Figure 12.*

| Label | Specifications |
|---|---|
| $CC_{1a}$ | $G = 8, N_i = N_j = N_k = N_l = 32, N_m = N_n = 16$ |
| $CC_{2a}$ | $G = 8, N_i = N_j = N_k = 32, N_m = N_o = N_p = 16$ |
| $CC_{3a}$ | $G = 8, N_i = N_j = 32, N_m = N_n = N_o = N_p = 16$ |
| $CC_{1b}$ | $G = 16, N_i = N_j = N_k = N_l = 16, N_m = N_n = 8$ |
| $CC_{2b}$ | $G = 16, N_i = N_j = N_k = 16, N_m = N_o = N_p = 8$ |
| $CC_{3b}$ | $G = 16, N_i = N_j = 16, N_m = N_n = N_o = N_p = 8$ |
| $MM_a$ | $G = 2, N = 10000$ |
| $MM_b$ | $G = 100, N = 2000$ |
| $MPS_a$ | $G = 2, N_i = N_k = N_m = 3000, N_j = 10, N_l = 1$ |
| $MPS_b$ | $G = 5, N_i = N_k = N_m = 700, N_j = 10, N_l = 1$ |
| $PEPS_a$ | $G = 2, N_i = N_j = 400, N_k = N_l = N_m = N_n = 20$ |
| $PEPS_b$ | $G = 10, N_i = N_j = 64, N_k = N_l = N_m = N_n = 8$ |

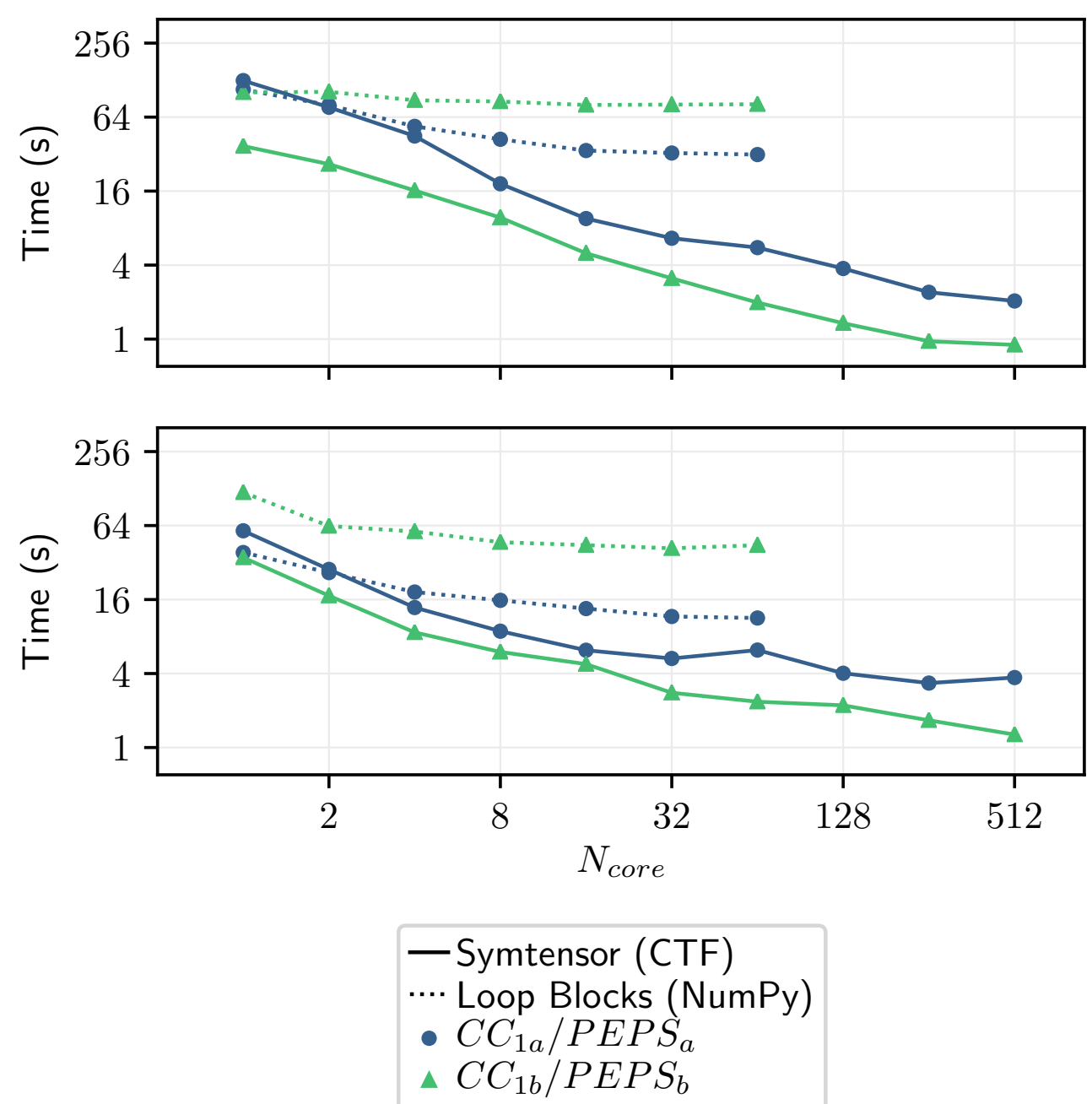

FIGURE 12. *Strong scaling across up to 8 nodes for the CC contractions (top) labelled $CC_{1a}$ (blue circles) and $CC_{1b}$ (green triangles) and the PEPS contractions (bottom) labelled $PEPS_a$ (blue circles) and $PEPS_b$ (green triangles). The dashed lines correspond to calculations done using a loop over blocks algorithm with a NumPy contraction backend while the solid lines correspond to Symtensor calculations using the irrep alignment algorithm, with a Cyclops contraction backend.*

when there are many symmetry sectors in each tensor. For example, using 64 threads on a single node, the speedup achieved by Symtensor over the loop over blocks implementation is 41X for $CC_{1a}$, 5.7X for $CC_{1b}$, 4.1X for $PEPS_a$ and 27X for $PEPS_b$. We additionally see that the contraction times continue to scale with good efficiency when the contraction is spread across multiple nodes.

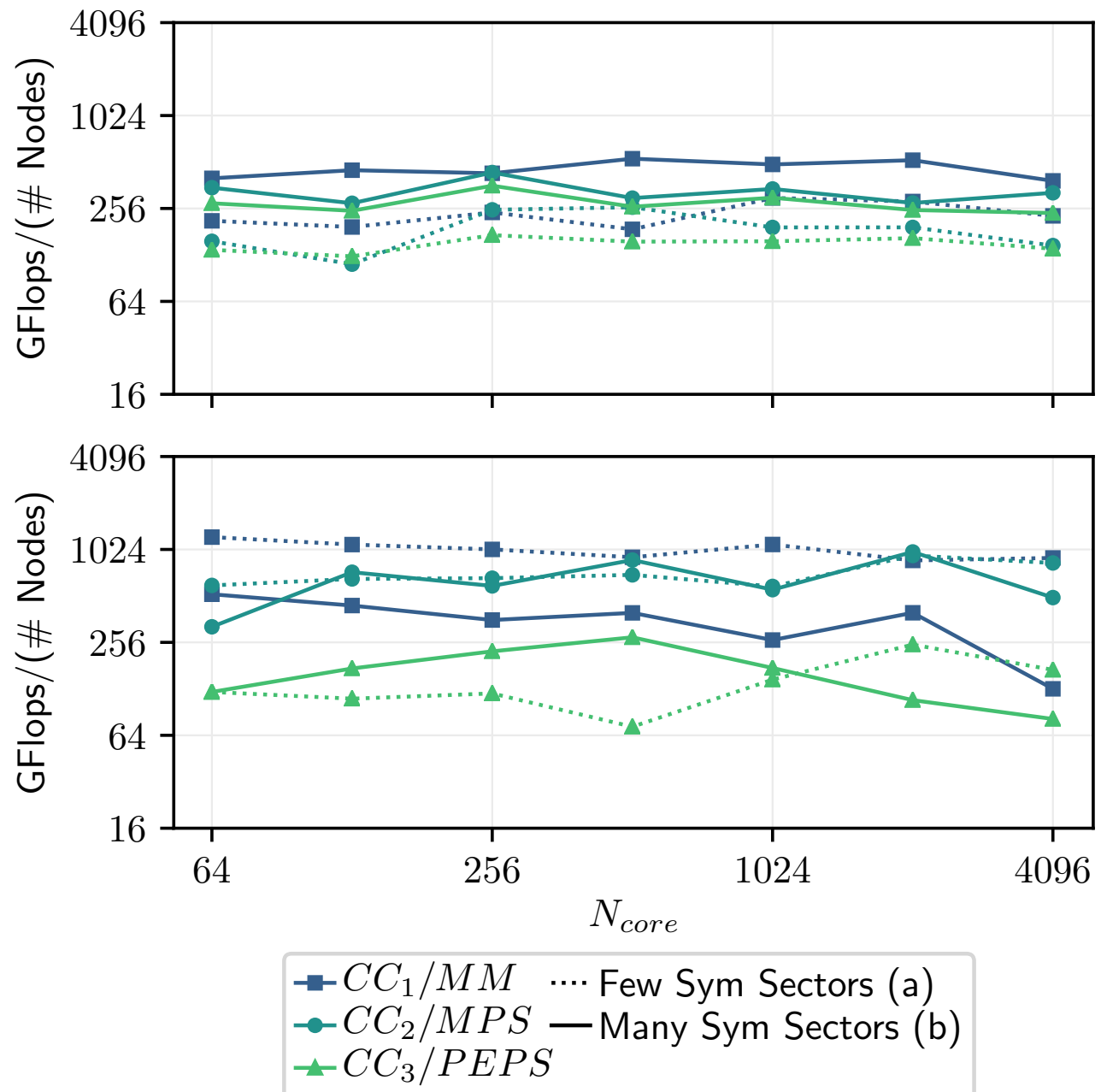

FIGURE 13. *Weak scaling (see text for details) across up to 64 nodes for CC (top) and TN (bottom) contractions, showing the performance, in GFlops per node, as a function of the number of used nodes. The dashed lines correspond to contractions with a small symmetry group (small G), previously labelled (a), while solid lines correspond to contractions with a large symmetry group (large G), labelled (b). The blue squares correspond to the $CC_1$ and matrix multiplication performance, the dark green circles correspond to the $CC_2$ and MPS performance, and the light green triangles correspond to the $CC_3$ and PEPS performance.*

Finally, in Figure 13 we display weak scaling performance, where the dimensions of each tensor are scaled with the number of nodes (starting with the problem size reported in Table 2 on 1 node) used so as to fix the tensor size per node. Thus, in this experiment, we utilize all available memory and seek to maximize performance rate. Figure 13 displays the performance rate per node, which varies somewhat across contractions and node counts, but generally does not fall off with increasing node count, demonstrating good weak scalability. When using $4096$ cores, the overall performance rate approaches 4 Teraflops/s for some contractions, but is lower in other contractions that have less arithmetic intensity.

**6. Conclusion.** The irrep alignment algorithm leverages symmetry conservation rules implicit in cyclic group symmetry to provide a contraction method that is efficient across a wide range of tensor contractions. This technique is applicable to many numerical methods for quantum-level modelling of physical systems that involve tensor contractions. The automatic handling of group symmetry with dense tensor contractions provided via the Symtensor library provides benefits in productivity, portability, and parallel scalability for such applications.

**7. Acknowledgement.** We thank Linjian Ma for providing the batched BLAS backend used in our calculations. ES was supported by the US NSF OAC SSI program, via awards No. 1931258 and No. 1931328. YG, PH, GKC were supported by the US NSF OAC SSI program, award No. 1931258. PH was also supported by a NSF Graduate Research Fellowship via grant DGE-1745301 and an ARCS Foundation Award. The work made use

of the Extreme Science and Engineering Discovery Environment (XSEDE), which is supported by US National Science Foundation grant number ACI-1548562. We use XSEDE to employ Stampede2 at the Texas Advanced Computing Center (TACC) through allocation TG-CCR180006.

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
