# Peer review of "Automatic transformation of irreducible representations for efficient contraction of tensors with cyclic group symmetry"

_SciPost Physics_

## Round 3 · Referee Report · Anonymous (Referee 1) · 2022-8-4

Strengths

1) relevant for many applications 2) available code repository 3) numerical benchmark

Weaknesses

1) abstract and rather technical 2) no concrete application to a physical problem performed 3) no documentation in the repository

Report

This manuscript describes an automatic transformation of cyclic group symmetries for efficient tensor contraction. As the authors pointed out, tensor based methods are frequently used in various fields of science. Many models exhibit different types of symmetries which induce sparsity of the tensors and simplify the problem. The proposed remapping induces a speedup compared to other sparse techniques (le.g., loops over symmetry blocks). Tensor contractions consume many resources and any optimization should be explored and utilized.

The algorithm has the potential for follow-up work as it can be integrated into other software applications. It would be helpful to add a detailed documentation to the repository.

The repository linked in SciPost does not work. I assume that the following repository is correct: https://github.com/yangcal/symtensor

The manuscript is well written but rather technical which makes it difficult for non-experts to understand the algorithms. It is only important for specialists.

Exploiting symmetries in tensor based methods is not a new idea and the automatic transformation is important but not a breakthrough result.

However, the manuscript involved substantial work that should not be unnoticed. It would be appreciated to demonstrate the speedup in a concrete algorithm for a simple physical problem (e.g. 1D or 2D Heisenberg model).

The manuscript is rather technical and lacks physical content at this stage. Currently, I cannot recommend it for publication in Scipost physics.

It might be a good contribution to SciPost Physics Codesbases if the authors could provide an easily accessible repository with detailed documentation.

Requested changes

Questions and comments:

1) Could the author (briefly) explain how to obtain the reduced form of a tensor for a specific problem? It would be useful to add some references.

2) Is it possible to exploit multiple cyclic group symmetries (translations in two dimensions?)

2) Regarding non-uniform blocks: how does the algorithm perform when the padding with zeros is not applied?

3) Does DPD have the same scaling advantages?

4) Why does Symtensor not use the NumPy backend as the other methods in Figure 10?

5) Please add 1/N_core line for comparison in Figure 12.

6) Could the authors (briefly) comment on the possibility to include non-abelian symmetries in a similar way?

7) The repository linked to SciPost should be fixed.

8) It would be appreciated to see a successful application of SymTensor to a real physical problem.

Small typos:

t1) Top paragraph on page 4: “as the standard reduced forms” -> “as the standard reduced form”

t2) Top paragraph on page 4: The authors write “This can be achieved first …”. Where is the “and second” associated with the “first”?

t3) Inconsistent style: Fig. vs. Figure

t4) Top paragraph on page 11: There is a space missing “tensors.In this” -> “tensors. In this”

---

## Round 3 · Referee Report · Johannes Hauschild (Referee 2) · 2022-8-9

Strengths

1) Clear and well written 2) Meets all the general aceptence criteria of Scipost Physics

Report

The general idea for the implementation of abelian symmetries in tensor contractions through block-sparse tensors is well known. In the manuscript, the authors present a reformulation in terms of dense higher-order tensors, applicable to cyclic group symmetries with uniform block sizes.
Further, they provide extensive benchmarks of their Symtensor library implementing this idea for different backends. They demonstrate an impressive, strong scaling over up to 8 nodes which is much harder to achieve with the traditional implementation method.

I support a publication. However, strictly speaking, I would not consider the idea "groundbreaking" enough to meet the strong expectations of Scipost Physics. Rather, I think the presented manuscript would fit better in the new Scipost Physics Codebases Journal, and would recommend a publication there. To meet that journal's criteria, the only things missing are a short ReadMe with installation instructions in the provided Github repository, and a little bit of code documentation (which would be helpful in any case).
On the other hand, the manuscript is rather focused on the general idea than the specific implementation, such that Scipost Physics Codebases might not be the perfect journal either. I leave it to the editor(s) to ultimately decide where the paper should be published.

Requested changes

1) In Algorithm 2.2, it's not clear (to me) what the two rows of the coefficient vectors are representing.

2) In Section 3, you write that the library also supports U(1) symmetries. However, in that case there's no reason to assume that the various symmetry blocks have equal sizes, while the presented algorithm assumes this. Can you please comment on that? You mentioned earlier that you just fill up blocks with explicit zeros, so do you significantly loose sparsity in this case?

3) Typos: - in the line before eq. (2.1): missing index k, and upper-case I (as India) instead of lower-case l (as Lima). - Page 12: contrac[i]ton

4) (optional) In Fig. 9, I would suggest to add "speedup" to the ylabel to make it clearer upon first read what is shown here.

---

## Round 3 · Referee Report · Juraj Hasik (Referee 3) · 2022-8-13

Strengths

1) Addresses one of the core problems of scaling tensor-based numerical methods beyond single node computations 2) Performance on a rich set of exemplary contractions is demonstrated, both in single and multi-node setting 3) The proposed solution is a thin layer on top of highly-optimized dense kernels 4) source code provided*

  • currently provided link "https://github.com/philliphelms/symtensor" does not point to a public repo. It is not clear, if the publicly available repo "https://github.com/yangcal/symtensor" contains the newest version of the library

Weaknesses

1) Lacks an end-to-end benchmark of a physical problem 2) the MPS and PEPS contraction examples between symtensor and "loop over blocks" can use realistic charge distribution 3) The source code is currently missing documentation

Report

From the perspective of numerical simulations in condensed matter theory, the invention of DMRG by White in 92' opened a new framework to address effective lattice models of materials. In the following three decades these methods, heavily relying on tensor algebra, proved to be extremely powerful: MPS are the reference method in 1D, while in 2D the tensor networks, i.e. PEPS or MPS on cylinders, are defining the state-of-the art together with Monte Carlo.

The gain further insight into challenging problems of condensed matter via tensor networks simulations, ability to reach even higher bond dimensions is called for. Contrary to Monte Carlo, efficiently utilizing modern super-computer architectures to that end remains a challenge.

The submitted manuscript is addressing this challenge. It presents a clever computational technique exploiting specific block-sparse structure of tensors with cyclic symmetry. Depending on the contraction, the blocks of tensor are re-indexed (and re-mapped in memory), such that efficient dense kernels can be utilized to perform the contractions of dense blocks. The benchmarks demonstrates very good scaling up to O(10^3) cores over O(10) nodes.

In its current form the manuscript presents no new physical results, rather it reports an advance in tensor-algebra of tensors with specific block-sparsity on modern super-computer architectures. Therefore, I do not recommend its publication in SciPost Physics, but it fits very well into the scope of SciPost Physics Codebases journal.

Questions: * More detailed discussion of U(1) symmetry, which typically arises in physics context, is missing. If U(1) is not treated approximately, i.e. by some finite (cyclic) subgroup, the number of differently charged blocks is unbounded. Typically, the size of the charged sectors (often labeled N in the manuscript) follows Gaussian distribution with respect to U(1) charge. How does the symtensor approach compare to "loop over block" approach in such case ?

  • The "loop over blocks" method, adopted widely in other libraries, in theory allows for parallelization of the loop since the contractions of individual blocks are independent. Do the comparisons in the manuscript use such parallelization of "loop over blocks" algorithm ?

  • Why do examples MM_b, MPS_a, MPS_b, PEPS_a, and PEPS_B in Fig. 11
    using BLAS perform slightly better than CTF ?

Minor remarks: * I believe beside ITensor and Uni10, the TenPy project is missing among the leading packages for abelian-symmetric tensor network computation frameworks (targeting MPS in this case)

  • page 12: I suppose "... boundary contracton approach [33]" should read "... contraction ..."

---

## Editorial Decision

resubmitted